# Completing Autophagy: Formation and Degradation of the Autophagic Body and Metabolite Salvage in Plants

**DOI:** 10.3390/ijms21062205

**Published:** 2020-03-23

**Authors:** Szymon Stefaniak, Łukasz Wojtyla, Małgorzata Pietrowska-Borek, Sławomir Borek

**Affiliations:** 1Department of Plant Physiology, Faculty of Biology, Adam Mickiewicz University Poznań, Uniwersytetu Poznańskiego 6, 61-614 Poznań, Poland; szymon.stefaniak@amu.edu.pl (S.S.); wojtylal@amu.edu.pl (Ł.W.); 2Department of Biochemistry and Biotechnology, Faculty of Agronomy and Bioengineering, Poznań University of Life Sciences, Dojazd 11, 60-632 Poznań, Poland; malgorzata.pietrowska-borek@up.poznan.pl

**Keywords:** Atg proteins, autophagosome, tonoplast, vacuole, SNARE proteins

## Abstract

Autophagy is an evolutionarily conserved process that occurs in yeast, plants, and animals. Despite many years of research, some aspects of autophagy are still not fully explained. This mostly concerns the final stages of autophagy, which have not received as much interest from the scientific community as the initial stages of this process. The final stages of autophagy that we take into consideration in this review include the formation and degradation of the autophagic bodies as well as the efflux of metabolites from the vacuole to the cytoplasm. The autophagic bodies are formed through the fusion of an autophagosome and vacuole during macroautophagy and by vacuolar membrane invagination or protrusion during microautophagy. Then they are rapidly degraded by vacuolar lytic enzymes, and products of the degradation are reused. In this paper, we summarize the available information on the trafficking of the autophagosome towards the vacuole, the fusion of the autophagosome with the vacuole, the formation and decomposition of autophagic bodies inside the vacuole, and the efflux of metabolites to the cytoplasm. Special attention is given to the formation and degradation of autophagic bodies and metabolite salvage in plant cells.

## 1. Introduction

Autophagy, which literally means “self-eating”, plays a crucial role in the degradation of useless or damaged cell components such as macromolecules, protein complexes, and organelles. Autophagy also plays a role in the degradation of foreign elements for cells, such as bacteria, viruses or sperm residues after egg cell fertilization. It is a conserved process that occurs in a similar way in fungal, animal and plant cells. This process was first observed in the 1960s, and for decades it was thought to involve non-selective degradation of cellular elements [1,2,3]. However, the results of studies performed in the last three decades reveal that autophagy is a highly advanced and specific process necessary for the proper functioning of the cell [4]. Under normal conditions, autophagy occurs at low intensity, but as a result of various abiotic and biotic stress factors (e.g., carbon or nitrogen starvation, salinity, drought, high temperature, reactive oxygen species or pathogens) the process is dramatically intensified [5,6,7,8,9,10,11]. In mammals, autophagy plays an important role during normal growth and development, starting from early embryogenesis [12]. It is important for maintaining good health because its efficient operation prevents the development of many diseases, including cancer, diseases of the liver, muscles, and heart, neurodegenerative diseases (e.g., Huntington’s disease), inflammation, pathogen infections, and aging [13,14,15,16,17,18,19]. In plants, autophagy participates in the circulation of cell components and acts as a quality control mechanism. It also functions in some developmental processes such as pollen maturation, aging, and cell death, including programmed cell death [6,7,20,21,22,23].

Some stages of autophagy are currently subject to intensive study, and our knowledge is gradually increasing. Areas of strong research interest include the initiation of autophagy, formation and elongation of the phagophore, and identification of the receptors and scaffold proteins involved in selective kinds of autophagy. However, despite many years of research, some aspects of autophagy are still not fully understood. Our knowledge about some stages of autophagy is poor, even fragmentary, and in addition, these stages are currently not of interest to many scientists. These include the final stages of autophagy, in particular, the degradation of autophagic bodies and the recovery of metabolites constituting the final products of autophagy. We specifically focus on the formation and degradation of autophagic bodies and metabolite salvage in plant cells and we compare this knowledge to data related to yeast.

## 2. Formation of the Autophagic Body during Macroautophagy

### 2.1. Formation and Trafficking of Autophagosome

Macroautophagy is by far the best-studied and described type of autophagy. The first visible symptom of macroautophagy is the appearance in the cytoplasm of a cup-shaped structure, called the phagophore (Figure 1). The phagophore elongates, surrounding and simultaneously separating the fragment of the cytoplasm together with organelles or other components of the cell that are intended for degradation. The final stage of phagophore differentiation is the complete surrounding of the cargo and its sequestration inside the autophagosome. This is a vesicle with a double, bilayer lipid-protein membrane, containing cargo intended for autophagic degradation [24,25,26,27,28,29]. The mechanisms of these initial stages have been intensively studied and described in numerous review papers, e.g., [30,31,32,33]. In yeast and plants, the autophagosome fuses with the vacuole creating an autophagic body that is quickly degraded by vacuolar lytic enzymes [34,35,36,37]. In animals, the autophagic body is not formed because autophagosome fuses with the lysosome, which delivers lytic enzymes enabling the degradation of the cargo inside the autolysosome (Figure 1) [29,38].

Components of the cytoskeleton play an important role in the cytoplasmic transport of autophagosomes [39]. It is also suggested that the cytoplasmic transport of autophagosomes is enabled by the microtubule network controlled by endosomal sorting complexes required for transport (ESCRT) [25]. The occurrence of ESCRT was evidenced in yeast [40,41] and plants [42,43]. An important role in this transport is played by fully developed autophagosomes with the outer-membrane-anchored Atg8 and phosphatidylinositol 3-phosphate (PI3P) in yeast (Figure 2, Table 1) and plants (Figure 3, Table 1) [44,45,46,47,48]. The protein FYCO1 (FYVE and coiled-coil domain-containing protein 1) is also important in the association of the autophagosome membrane and microtubules, and it has a modular structure composed of four amino acid domains and spiral signaling domain FYVE [46,49]. Due to its structure, FYCO1 interacts with the autophagosome surface simultaneously in two places—with Atg8 and PI3P (both in yeast and plants). These two sites for recognition and linking of the autophagosome with the FYCO1 make it possible to distinguish between mature autophagosomes and phagophores [46,50,51] because only on the surface of the mature autophagosomes are there simultaneously proteins necessary to form stable and double bonds with FYCO1. Attached to the surface of the autophagosome, FYCO1 also binds to GTP-binding protein 7 (Ypt7) in yeast and Ras-related protein RAB7 (RAB7) in plants [49,52,53], creating an autophagosome-FYCO1-Ypt7/RAB7 system that allows the binding of the autophagosome to microtubules. The autophagosome-FYCO1-Ypt7/RAB7 system moves to the plus end of microtubules by the binding of Ypt7/RAB7 to the kinesin motor proteins [52,54].

### 2.2. Fusion of the Autophagosome with the Vacuole and Formation of the Autophagic Body

During the fusion of the autophagosome with the vacuole in yeast, the outer membrane of the autophagosome is connected and incarnated to the tonoplast, while the inner membrane together with the content becomes the autophagic body inside the vacuole (Figure 1, Figure 2) [24,25,55,56]. In yeast, during fusion, protein Ypt7 and complex HOPS are involved (Figure 2, Table 1) [57,58]. Ypt7 is involved in autophagy in *Magnaporthe oryzae*, *Pichia pastoris*, *Saccharomyces cerevisiae*, and *Schizosaccharomyces pombe* [59,60,61,62,63,64,65]. Ypt7 is recruited to the surface of the autophagosome by the interaction with the calcium caffeine zinc sensitivity 1-monensin sensitivity 1 complex (Ccz1-Mon1), which is the Ypt7 guanine nucleotide exchange factor. Furthermore, in the fusion of the autophagosome and tonoplast in yeast such proteins as cis-Golgi membrane traffic (Vti1) [66], syntaxin VAM3 (Vam3) [67], syntaxin VAM7 (Vam7) [68,69,70], and Ykt6 are involved (Figure 2, Table 1) [58,71,72,73].

In plants, the fusion of autophagosome and vacuole and the mechanisms regulating this process are poorly understood. So far, only the involvement of protein VTI12 has been confirmed in the fusion of autophagosome and vacuole in plants (Figure 3, Table 1). This protein belongs to the complex named soluble N-ethylmaleimide-sensitive factor activating protein receptors (SNARE proteins). Plant VTI12 is the ortholog of yeast Vti1 (Figure 2, Table 1). In *Arabidopsis* mutants with T-DNA VTI12 insertion, growing in rich-nutrient conditions, presented a normal phenotype, whereas under nutrient-poor conditions accelerated aging was observed, confirming that VTI12 is involved in autophagy in plants [25,74,75,76,77]. This protein also participates in the transport of storage proteins from the cytoplasm to the vacuole [76]. Although plants express many SNARE proteins that are involved in a variety of processes such as defense against pathogen attack [78] or intracellular transport [79], VTI12 is the only SNARE protein that has been proven to be involved in the fusion of the autophagosome and vacuole in plants (Figure 3, Table 1). Other protein that may be involved in the fusion of autophagosome and vacuole in plants is RABG3B (Figure 3, Table 1). So far, the occurrence of RABG3B has been confirmed in *Arabidopsis* and *Populus* and it participates in, among other processes, the formation of the wood conductive elements when programmed cell death occurs [80,81]. The protein RABG3B is located at the surface of the autophagosome, however, it remains unclear whether RABG3B can regulate the fusion of the autophagosome and vacuole in plants. It is suggested that the homologous yeast proteins such as Ykt6, Vam3, Ypt7, and complex HOPS are involved in the fusion of autophagosome and vacuole in plants [46,82]. Moreover, it is also suggested that the plant components of the ESCRT complex, such as the charged multi-vesicular body protein 1 (CHMP1), FYVE-domain protein required for endosomal sorting 1 (FREE1), vacuolar protein sorting 2.1 (VPS2.1), cell death-related endosomal FYVE/SYLF protein 1 (CFS1), and plant exocyst complex component EXO70B1 (EXO70B1) are involved in trafficking of the autophagosome, the fusion of autophagosome and vacuole, and the release of the autophagic body into the vacuole [25]. Nevertheless, the detailed localization and function of these proteins and complexes are not known.

## 3. Formation of the Autophagic Body during Microautophagy

The amount of data in the literature that describes the process of microautophagy in fungi, animals and plants, including its course, regulation mechanisms, and importance for the cell, is much smaller than the information on macroautophagy. During microautophagy, an autophagosome is not formed, but the tonoplast creates an invagination into which the cargo moves. The invagination of the tonoplast increases and the cargo is engulfed into the vacuole forming an autophagic body (Figure 4) [83,84,85]. Microautophagy can also occur through the formation of an arm-shaped protrusion of the tonoplast when a portion of the cytoplasm is captured into the vacuole (Figure 4). So far, the formation of the protrusion has been confirmed for microautophagic degradation of peroxisomes in *Pichia pastoris* and degradation of anthocyanin aggregates in *Arabidopsis thaliana* [86,87]. However, the best-known mechanism of microautophagy is the absorption of the cargo into the vacuole by membrane invagination. It occurs as a result of changes in the organization of the structure of the vacuolar membrane, mainly through changes in the content of lipids and large transmembrane proteins [88]. The membrane is invaginated, and the speed and extent of these changes are regulated in yeast by Vps1p [89] (Table 2). Environmental factors, such as carbon or nitrogen starvation, also have a significant influence on the process of membrane differentiation, formation of an invagination, and the subsequent formation of an autophagic tube and autophagic body [88,90]. The formation of the autophagic tube in yeast is an ATP-dependent process [91]. Moreover, the processes of membrane differentiation and formation of the autophagic tube and autophagic body are regulated by numerous Atg proteins and signaling complexes. Two Atg7-dependent ubiquitin-like conjugation systems (UBLC) are involved in the regulation of microautophagy. The first of these consists of Atg8 coupled with phosphatidylethanolamine (Atg8-PE) by Atg7 as an E1-like enzyme and Atg3 as an E2-like enzyme. Moreover, in this UBLC system, cysteine protease Atg4 proteolytically removes the C-terminal of Atg8 [92,93]. The second UBLC system includes Atg12 covalently linked to Atg5 through a ubiquitin-dependent conjugation system consisting of Atg7 as an E1-like enzyme, and Atg10 as an E2-like enzyme. The Atg12-Atg5 dimer oligomerizes with Atg16 to stimulate the formation of the Atg8-PE complex [94,95]. Furthermore, the Atg7-dependent UBLC system, called the vacuolar transporter chaperone (VTC) complex, plays an important role in the formation of the autophagic tube in yeast by controlling the distribution of proteins in different regions of the membrane [89,96]. In addition, the VTC complex is a potential site for calmodulin activation and thus initiation of membrane invagination. Calmodulin is a factor stimulating tonoplast invagination during microautophagy and the formation of the autophagic tube. The top of the autophagic tube expands to form a pre-vesicular structure. The autophagic tube and the pre-vesicular structure are formed based on sorting mechanisms and differences in the density of proteins and lipids occurring in the structure of the autophagic tube. The newly formed vesicles expand due to the action of E1 and E2-like enzymes, and then bud from the end of the tube, forming the autophagic body inside the vacuole [89,97].

Microautophagy, similarly to macroautophagy, can occur in a selective manner. One example of selective microautophagy is micropexophagy—an autophagic degradation of peroxisomes [98]. During micropexophagy peroxisomes are surrounded by an arm-shaped protrusion of vacuole membrane, the vacuolar sequestering membranes (VSM). It is suggested that the key structure involved in the formation of VSM is a perivacuolar structure (PVS), which is a structure similar to the pre-autophagosomal structure (PAS) in yeast macroautophagy. Moreover, PVS participates in the formation of the micropexophagy-specific membrane apparatus (MIPA) [99]. MIPA is also a key structure necessary for the complete surrounding peroxisomes intended for degradation during micropexophagy. MIPA is a double-membrane, small, cup-shaped structure whose fusion with VSM closes the peroxisomes inside the vacuole [100,101,102]. Inside the vacuole, a single-membrane vesicle containing peroxisomes is released, and the forming autophagic body is called a micropexophagic body. In contrast to macroautophagy, proteins Atg1, Atg11, Atg26, Atg28, Atg30, and Atg35 are necessary for microautophagy to occur in *Pichia pastoris*. It has been suggested that these proteins play a role in the early stages of microautophagy such as recognition and mobilization of peroxisomes intended for autophagic degradation [102,103]. Proteins Atg3, Atg4, Atg7, Atg8, Atg26, Atg28, and Atg30 are involved in the formation of MIPA [100,103,104,105,106]. Proteins specific for VSM formation during micropexophagy are Atg11, Atg17, Atg30, and Atg37 [105,107] (Table 2). Another type of selective microautophagy is micromitophagy, i.e., autophagic degradation of mitochondria. Micromitophagy occurs by transferring mitochondria from the cytoplasm to the lumen of the vacuole by invagination of the vacuolar membrane [88,90,108,109]. The increasing invagination of the vacuolar membrane engulfs the mitochondria intended for degradation and forms an autophagic tube. At the end of the autophagic tube, the mitochondria-containing autophagic body is formed. Two UBLC systems, kinase-Atg1, Atg9, Atg11, Vac8, and Vam7 are involved in micromitophagy in yeast [108,110] (Table 2). In addition to micropexophagy and micromitophagy, the piecemeal microautophagy of the nucleus (PMN) is described as a selective type of microautophagy. This is the autophagic degradation of a part of the cell nucleus. This process is initiated by the fusion of membranes of the cell nucleus and vacuole. The tonoplast protein Vac8 and protein Nvj1, being a part of the nuclear envelope, are involved in the fusion [111] (Table 2). The next stage is the formation of tonoplast invagination that increases and transforms into an intra-vacuolar vesicle consisting of three membranes (tonoplast and two nuclear envelope membranes) and a portion of the nucleoplasm. After the formation of the intra-vacuolar vesicle, the cell nucleus and vacuole separate and the autophagic body is formed. Effective degradation of nuclear elements during PMN requires the expression of core *Atg* genes [112,113]. However, the participation of Atg proteins seems to be limited to the final stages of this process, namely, the stage of the detachment of a fragment of the cell nucleus and closing of the tonoplast [114].

## 4. Degradation of the Autophagic Body and Metabolite Efflux from the Vacuole to the Cytoplasm

Only a few publications describe the degradation of the autophagic body and is often written about using generalities, predictions, and suggestions. The degradation is rapid and begins immediately after its appearance inside the vacuole. Compared to the initial stages of autophagy, the mechanism and regulation of the degradation of the autophagic body and the efflux of metabolites from the vacuole to the cytoplasm are very poorly investigated and understood. However, these are key stages on the path to recycling cellular components in the entire autophagy process.

Proteins involved in the degradation of the autophagic body in yeast are proteinase A (Pep4) and proteinase B (Prb1) (Figure 2, Table 3). These proteins are involved in the activation of cascades of other vacuolar proteases and hydrolases, which are indicated as key factors involved in the degradation of the autophagic body in yeast [34,115,116]. However, the cascades involved in autophagic body degradation have not yet been identified in detail. The best-known and described protein involved in the degradation of the autophagic body in yeast is Atg15, a putative lipase (Figure 2, Table 3). It has been proven that this protein plays a role in the degradation of the autophagic body, not only through the decomposition and recycling of components of the autophagic body membrane but also due to it being a key protein involved in the degradation of the cargo located inside the autophagic body [34,117,118,119,120]. In addition to those described above, there are several other proteins that are thought to be involved in the degradation of the autophagic body in yeast, but their mechanism of action and functions are not yet fully understood. Proteins whose participation in the degradation of the autophagic body has not yet been undeniably confirmed are Atg22, Atg42, Ybr139, and Prc [121,122,123] (Figure 2, Table 3). After the degradation of the autophagic body, the metabolites must be transported to the cytoplasm. Unfortunately, there is minimal knowledge available about this stage of autophagy. It is thought that the putative vacuolar permease Atg22 is the protein that may be responsible for this transport. This protein may cooperate with two other vacuolar permeases Avt3 and Avt4 [123,124,125], which mediate the transport of leucine and other amino acids from the vacuole to the cytoplasm [123,124,125,126]. Nevertheless, the involvement of Avt3 and Avt4 in the transport of amino acids derived from the degradation of the autophagic body during autophagy in yeast is still hypothetical (Figure 2, Table 3) [123,124,125].

So far, in plants, the events occurring during and after the degradation of the autophagic body inside the vacuole are poorly understood. Only one protein has been described in plants that is likely to be involved in the degradation of the autophagy body. The VPEγ protein described in *Arabidopsis thaliana* is likely to act similarly to yeast Pep4 by activating cascades of other hydrolases that are responsible for the hydrolysis of various structures found inside vacuoles, including the autophagic body (Figure 3, Table 3) [127]. However, these hydrolases are not precisely indicated. In the literature, there are no reports of plant proteins being involved in the efflux of metabolites released during the degradation of the autophagic body from the vacuole to the cytoplasm. Only the existence of vacuolar permease AVT3 was confirmed in *Arabidopsis thaliana* [128], but it is not known how important this permease is in the transport of metabolites coming from the degradation of the autophagic body in plants.

## 5. Regulation of Autophagic Body Degradation

The regulation of the whole autophagy process is an extensive topic, which has been intensively studied for several decades. In short, autophagy under normal conditions occurs at a low intensity; however, this process is clearly enhanced as a result of various types of abiotic and biotic stresses [29,129,130,131]. In yeast, the main factors that increase the intensity of autophagy are carbon, nitrogen and phosphate starvation [24,129]. In plants, it is known that carbon or nitrogen starvation significantly increases the intensity of autophagy [7,24,29,36,132,133]. In yeast, animal and plant cells, mTOR kinase is the main intracellular center of signal collection associated with autophagy. Amino acids are involved in the activation of mTOR kinase, which apart from autophagy regulates such processes as growth, proliferation, cell movement, and protein translation [134,135,136]. It has also been proven that autophagy plays a key role in maintaining the level of free amino acids in the cell and protein synthesis under stress [137], but it has not been explained how amino acids as one of the end products of autophagy regulate this process. Furthermore, in contrast to the well-known mechanisms regulating autophagy in general, surprisingly little is known about the regulation of the degradation of the autophagic body, which is one of the final stages of autophagy. So far in plants, it has only been observed that the degradation of the autophagic body is clearly inhibited by asparagine during sugar starvation-induced autophagy in cells of lupin (*Lupinus* spp.) embryo axes during seed germination [133]. Enhanced autophagy was found in sugar-starved cells of lupin embryo axes, which was evidenced, among other things, by a high degree of vacuolization and a clear decrease in phosphatidylcholine content [133,138,139]. Under such conditions, the autophagic bodies were rapidly degraded and they were not observed in the enlarged vacuoles. Nevertheless, asparagine, a central amino acid in the metabolism of germinating protein lupin seeds [140], caused clear inhibition in the degradation of autophagic bodies and their accumulation in highly enlarged vacuoles. Such accumulation of autophagic bodies inside enlarged vacuoles proves that asparagine only inhibits the degradation of autophagic bodies, but does not reduce the intensity of autophagy in general. The effect of asparagine is similar to the action of concanamycin A (an inhibitor that slows down the breakdown of the autophagic body by lowering the pH of the vacuole); however, the mechanism of the inhibitory action of asparagine is not known [133].

## 6. Conclusions and Future Perspectives

Research on autophagy is currently being carried out in many research centers around the world and is the focus of interest for many research teams. The research on autophagy has two faces. On one hand, the initial stages of autophagy, selective types of autophagy and its significance in the etiology, course, and prevention of diseases is being studied intensively, while on the other hand, there are still many aspects of autophagy that are not popular among the scientific community. The events occurring in the final stages of autophagy have been explored only marginally and require a lot of research to be fully understood. Knowledge regarding the formation and, in particular, degradation of the autophagic body is sparse. Also, the efflux of metabolites from the vacuole to the cytoplasm is poorly investigated and understood. These stages of autophagy have not been intensively studied, indeed, they have been analyzed somewhat incidentally and are often written about using generalities and conjecture. Nevertheless, these latter stages are very important stages, because they complete the entire process of autophagy. Although autophagy has been known since the 1960s, and in the last two or three decades our knowledge on autophagy has increased dramatically, there are still many unanswered questions. For example, is the autophagic body degraded by nonspecific vacuolar lytic enzymes, or is the autophagic body degraded by some specific autophagy-related enzymes? Or do both of these enzymes participate in the degradation of the autophagic body? Another poorly understood stage of autophagy in plants is the transport of metabolites from the vacuole to the cytoplasm after autophagic body degradation. Also, in this case, there are other unanswered questions such as whether the constitutive vacuolar membrane transporters are involved in the transport of metabolites to the cytoplasm or whether some autophagy-related transporters are necessary for the metabolite salvage at the end of autophagy. It is also unknown whether, and how the spectrum and level of vacuolar lytic enzymes and membrane transporters change during enhanced autophagy occurring under different stresses, for example, during carbon or nitrogen starvation. In summary, so far, we have only discovered the tip of the iceberg and there remains much to be explored on the way to a full understanding of the whole process of autophagy.

## Figures and Tables

**Figure 1 ijms-21-02205-f001:**
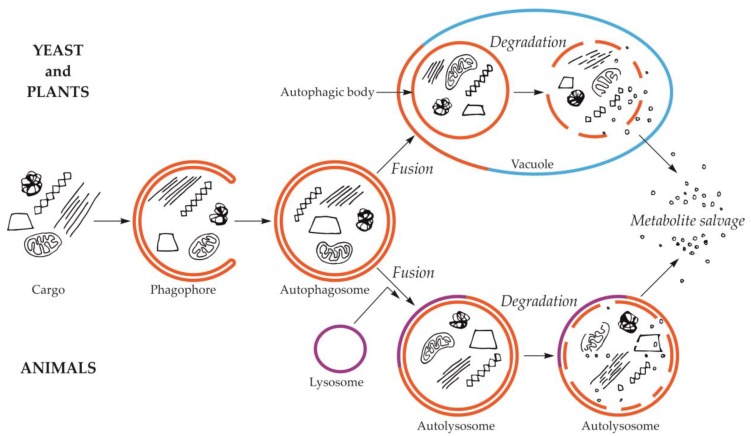
Schematic diagram of macroautophagy in cells of yeast and plants (upper part of the drawing) and in cells of animals (bottom part of the drawing). In yeast and plant cells, the autophagosome fuses to the tonoplast, creating the autophagic body inside the vacuole. In animal cells, the autophagosome fuses with the lysosome, giving the autolysosome. The autophagic body inside the vacuole and the content of autolysosome are rapidly degraded, allowing reuse of metabolites.

**Figure 2 ijms-21-02205-f002:**
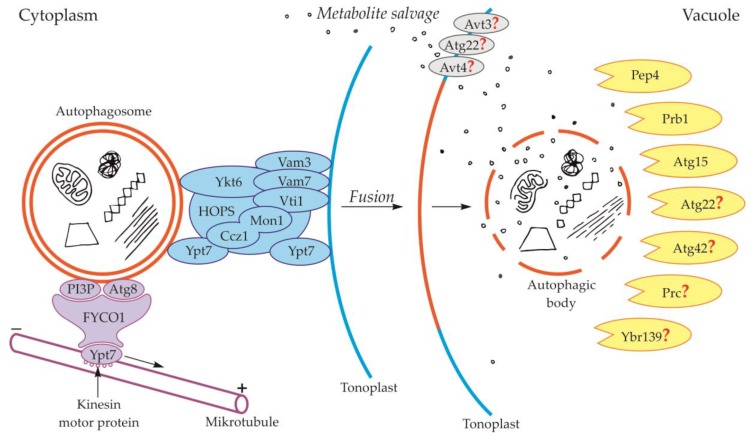
Schematic diagram depicting trafficking and fusion of the autophagosome to the vacuole and degradation of the autophagic body inside the vacuole in yeast. PI3P and Atg8 anchored in the outer membrane of the autophagosome are involved in autophagosome trafficking and bind autophagosome with FYVE and coiled-coil domain-containing protein 1 (FYCO1). The complex autophagosome-FYCO1-Ypt7 moves along microtubules in the direction of the plus end by the binding of Ypt7 to kinesin motor proteins. Proteins Vam3, Vam7, Vti1, Ykt6, Ypt7, and complexes Ccz1-Mon1 and HOPS are involved in the fusion of the autophagosome and vacuole. The newly formed autophagic body inside the vacuole is rapidly degraded by lytic enzymes. Proteins involved in the degradation of the autophagic body are proteinase A (Pep4), proteinase B (Prb1), and putative lipase Atg15. Other proteins that are probably involved in the degradation of the autophagic body are Atg22, Atg42, Prc, and Ybr139. It is suggested that Atg22, Avt3, and Avt4 are involved in metabolite efflux from the vacuole to the cytoplasm. Question marks indicate the hypothetical involvement of proteins in the degradation of the autophagic body and metabolite efflux from the vacuole to the cytoplasm.

**Figure 3 ijms-21-02205-f003:**
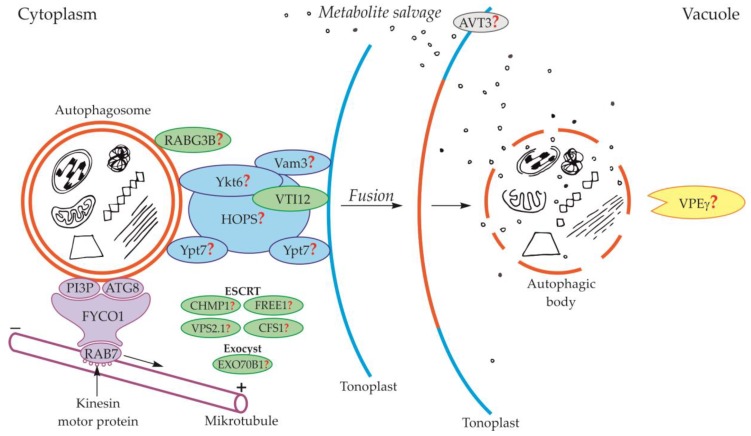
Schematic diagram depicting trafficking and fusion of the autophagosome to the vacuole and the degradation of the autophagic body inside the vacuole in plants. PI3P and ATG8 anchored in the outer membrane of the autophagosome are involved in autophagosome trafficking and bind autophagosome with FYCO1 protein. The complex autophagosome-FYCO1-RAB7 moves along microtubules in the direction of the plus end by the binding of RAB7 to kinesin motor proteins. Protein VTI12 is involved in the fusion of the autophagosome and vacuole. RABG3B is located on the surface of the autophagosome but the involvement of this protein in the fusion of the autophagosome and vacuole in plants remains unclear. It is suggested that the homologous yeast proteins Ykt6, Vam3, Ypt7, and complex HOPS are involved in the fusion of autophagosome and vacuole in plants. Additionally, it is suggested that plant proteins CHMP1, FREE1, VPS2.1, CFS1, and the complex EXO70B1 are involved in the autophagosome trafficking, autophagosome-vacuole fusion, and the release of the autophagic body into the vacuole lumen. The newly formed autophagic body inside the vacuole is rapidly degraded by lytic enzymes. One of them can be the vacuolar processing enzyme γ (VPEγ). Proteins involved in metabolite efflux from the vacuole to the cytoplasm during autophagy in plants have not been described so far. Only permease AVT3 was confirmed in *Arabidopsis thaliana*, but the involvement of this permease in the transport of metabolites coming from the degradation of the autophagic body is not confirmed. Question marks indicate the hypothetical involvement of plant proteins and complexes, or plant homologs of yeast proteins, during autophagy.

**Figure 4 ijms-21-02205-f004:**
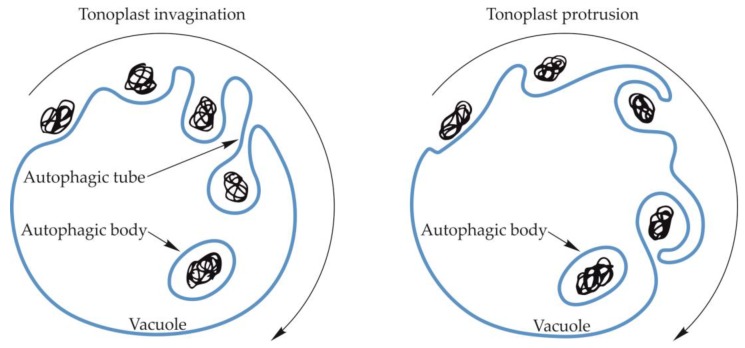
Schematic diagram of microautophagy in yeast and plants. Microautophagy may occur by vacuolar membrane invagination, which increases, creating the autophagic tube in yeast, and the cargo is enclosed inside the vacuole, forming a vesicle surrounded by a single two-layer membrane—an autophagic body. The occurrence of the autophagic tube in plants has not been confirmed. The autophagic body may also be formed by an arm-shaped protrusion of the tonoplast capturing a cargo into the vacuole.

**Table 1 ijms-21-02205-t001:** Proteins involved, or hypothetically involved, in trafficking and fusion of the autophagosome to the vacuole and formation of the autophagic body during macroautophagy.

Protein	Organism	Function	References
PI3P, Atg8	yeast, plant	autophagosome trafficking and fusion	[46,47]
Vti1	yeast	autophagosome formation, autophagosome-vacuole fusion	[66]
Ykt6	yeast	retrograde transport, vacuole homotypic fusion, vesicles and vacuole fusion	[71,72,73]
Vam3	yeast	endosome-autophagosome fusion, autophagosome maturation, autophagosome-vacuole docking and fusion	[66,67,70]
Vam7	yeast	autophagosome-vacuole fusion	[68,69,70]
Ccz1-Mon1, Ypt7	yeast	autophagosome-vacuole fusion	[59,63]
HOPS	yeast	autophagosome-vacuole fusion	[57,58]
	plant	probably autophagosome-vacuole fusion	[46,82]
VTI12	plant	probably autophagosome formation, docking, and autophagosome-vacuole fusion, storage protein transport from cytoplasm to vacuole	[74,75,76,77]
RABG3B	plant	autophagy enhancement during xylem development and pathogen-induced cell death, probably autophagosome formation and autophagosome-vacuole fusion	[78,79]
CHMP1, FREE1, VPS2.1, CFS1, EXO70B1	plant	probably autophagic trafficking, autophagosome-vacuole fusion, release of autophagic body	[25]

**Table 2 ijms-21-02205-t002:** Proteins involved in the formation of the autophagic body during microautophagy.

Protein	Organism	Function	References
Vps1p	yeast	regulation of vacuole membrane invagination	[89]
Atg12, Atg5, Atg10, Atg16	yeast	differentiation of vacuole membrane, formation of autophagic tube, autophagic body formation	[92,93,94,95]
Atg3, Atg4, Atg7, Atg8, Atg26, Atg28, Atg30	yeast	formation of the micropexophagy-specific membrane apparatus (MIPA) during micropexophagy	[100,103,104,105,106]
Atg11, Atg17, Atg37	yeast	formation of the vacuolar sequestering membranes during micropexophagy	[102,105]
Vac8, Nvj1	yeast	fusion of cell nucleus and vacuole	[111]

**Table 3 ijms-21-02205-t003:** Proteins involved, or hypothetically involved, in the degradation of the autophagic body and metabolite efflux from the vacuole to the cytoplasm during macroautophagy.

Protein	Organism	Function	References
Pep4 (Proteinase A), Prb1 (Proteinase B)	yeast	activation of protease and hydrolase cascades by proteolytic processing	[34,115,116]
Atg15,	yeast	degradation of autophagic body, decomposition and recycling of autophagic body membrane, proaminopeptidase I maturation	[34,117,118,119,120]
Atg22	yeast	tonoplast protein with limited homology to permeases, putative vacuolar transporter involved in the efflux of metabolites from the vacuole to the cytoplasm	[122,123]
Atg42, Ybr139, Prc1	yeast	probably degradation of the autophagic body	[56]
Avt3, Avt4	yeast	vacuolar efflux transporters potentially involved in the efflux of leucine and other amino acids derived from the degradation of the autophagic body	[123,124,125,126]
VPEγ	*Arabidopsis thaliana*	probably autocatalytically converted into a smaller active form, which, like yeast’s Pep4, might be involved in proteolytically downstream processes that are responsible for the degradation of various vacuolar constituents	[127]
AVT3	*Arabidopsis thaliana*	vacuolar efflux transporter potentially involved in the efflux of metabolites derived from the degradation of the autophagic body	[128]

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
