# Peer review of "Completing Autophagy: Formation and Degradation of the Autophagic Body and Metabolite Salvage in Plants"

_ijms, 2020, doi:10.3390/ijms21062205_

Round 1

Reviewer 1 Report

This is a splendid review which convincingly summarizes the importance of autophagy. The authors have done a great job in giving a clear picture of a very complex process. The only minor recommendation I have is in going from macrophagy to microphagy which is a bit too sudden and perhaps could be prepared better. However overall this is a very fine article.

Author Response

This is a splendid review which convincingly summarizes the importance of autophagy. The authors have done a great job in giving a clear picture of a very complex process. The only minor recommendation I have is in going from macrophagy to microphagy which is a bit too sudden and perhaps could be prepared better. However overall this is a very fine article.

We described macroautophagy in section two and microautophagy in section three. Unfortunately, section three was started just after Figure 3 and Table 1, and probably it could cause the impression that the transition between macro- and microautophagy is too sudden. To avoid such an impression, we moved Table 1 just after Figure 2 in subsection 2.1. In this way, now section three starts after the caption of Figure 3, not right after a table. We would like not to add any new sentences at the beginning of section three. The beginning of this section already is general, and in our opinion, it is enough for highlighting a new topic.

Reviewer 2 Report

Basically, this manuscript represents a review on autophagy with emphasis in plants.  It features a focus on the formation and degradation of autophagic bodies and metabolite salvage.  In general I find the review interesting, easy to read and likely useful.  I do have three specific reservations/suggestions. 

(a)  The abstract is rather long and is largely repeated as an introduction and then as the conclusions.  These sections could be shortened and better focused for a more effective presentation.

(b)  With a repeated call for more study on degradation and salvage this manuscript sounds more like a research proposal then a balanced review.  Again this is particularly evident in the three summary sections.

(c) Lastly, there are no literature references for any of the figures; are they entirely original as is suggested?

Author Response

Basically, this manuscript represents a review on autophagy with emphasis in plants. It features a focus on the formation and degradation of autophagic bodies and metabolite salvage. In general I find the review interesting, easy to read and likely useful. I do have three specific reservations/suggestions.

(a) The abstract is rather long and is largely repeated as an introduction and then as the conclusions. These sections could be shortened and better focused for a more effective presentation.

We shortened the abstract by removing three fragments. It was 270 words and now is 189 words. In this way, the abstract is as informative as it was, and it is less similar to the Introduction and the Conclusions.

(b) With a repeated call for more study on degradation and salvage this manuscript sounds more like a research proposal then a balanced review. Again this is particularly evident in the three summary sections.

We deleted some sentences from the text. They are sentences in lines 29-31 (Abstract) and lines 61-64 (Introduction). We would like not to remove ‘a research proposal’ from the last section of the manuscript. This section is entitled “Conclusions and Future Perspectives”, so such suggestions of new research are, in our opinion, appropriate in this place.

Additionally, we removed two words from the Conclusions, which should be not in this place (line 359/360 and 363).

(c) Lastly, there are no literature references for any of the figures; are they entirely original as is suggested?

All figures were designed and drew exclusively by the authors of the manuscript based on knowledge obtained from the scientific articles cited in the manuscript. None of the figures is copied or modified from other papers(s); thus, none references are needed below the figures in our manuscript.